# Data Synthesis for Alfalfa Biomass Yield Estimation

Jonathan Vance [1], Khaled Rasheed [1,2,*], Ali Missaoui [3] and Frederick W. Maier [2]

1   School of Computing, University of Georgia, 415 Boyd Graduate Studies, 200 D. W. Brooks Drive, Athens, GA 30602, USA
2   Institute for Artificial Intelligence, University of Georgia, 515 Boyd Graduate Studies, 200 D. W. Brooks Drive, Athens, GA 30602, USA
3   Department of Crop and Soil Sciences, Institute of Plant Breeding Genetics and Genomics, University of Georgia, 4317 Miller Plant Science, Athens, GA 30602, USA
*   Correspondence: khaled@uga.edu

**Abstract:** Alfalfa is critical to global food security, and its data is abundant in the U.S. nationally, but often scarce locally, limiting the potential performance of machine learning (ML) models in predicting alfalfa biomass yields. Training ML models on local-only data results in very low estimation accuracy when the datasets are very small. Therefore, we explore synthesizing non-local data to estimate biomass yields labeled as high, medium, or low. One option to remedy scarce local data is to train models using non-local data; however, this only works about as well as using local data. Therefore, we propose a novel pipeline that trains models using data synthesized from non-local data to estimate local crop yields. Our pipeline, synthesized non-local training (SNLT pronounced like sunlight), achieves a gain of 42.9% accuracy over the best results from regular non-local and local training on our very small target dataset. This pipeline produced the highest accuracy of 85.7% with a decision tree classifier. From these results, we conclude that SNLT can be a useful tool in helping to estimate crop yields with ML. Furthermore, we propose a software application called Predict Your CropS (PYCS pronounced like Pisces) designed to help farmers and researchers estimate and predict crop yields based on pretrained models.

**Keywords:** machine learning; data synthesis; generative models; alfalfa; biomass; precision agriculture; classification; climate change; yield prediction; deep learning

## 1. Introduction

The alfalfa crop is an important livestock feed and is crucial to global food security. In previous work, we used climate data to estimate alfalfa biomass yields. We compared the accuracies of feature selection techniques and machine learning (ML) models for this task. We obtained promising results using local training data with $R^2$ values over 0.90, as we had access to rich curated datasets from state university variety trials [1]. However, since our team is developing a software application to aid real-world farmers, whose datasets may be much smaller, the current work addresses the problem of estimating yields for very small target datasets. We find that local training on very small target datasets results in very low accuracy, while, non-local training on much larger datasets performs only about as well as local training. Our solution combines ideas inspired by [2], which shows success using pretrained models and sparse datasets, with ideas inspired by [3,4], which show the promise of deep learning generative models like the adversarial autoencoder (AAE) [3] and generative adversarial networks (GANs) [4]. We propose a novel pipeline where models are trained with data generated or synthesized by other deep learning (DL) models. In this pipeline, the synthesized training data are synthesized from non-local sources, and the resulting classifiers estimate local targets. We call this synthesized non-local training (SNLT pronounced like sunlight), and we show it consistently achieves better accuracy than both local and non-local training. We extend the work of Xu et al. [5] by using their conditional

tabular GAN (CTGAN) and tabular variational autoencoder (TVAE) synthesizers in our pipeline. The highest accuracy we obtained from SNLT was 85.7% using a decision tree classifier (DT) with CTGAN. We also obtained good results using data synthesized by TVAE, especially when training extreme gradient boosting models (XGBoost), scoring as high as 75.0% and 70.0% classification accuracy. One long-term goal of this work is to develop a software application called Predict Your CropS (PYCS, pronounced like Pisces) [6]. PYCS is a research software application and what-if tool that farmers and others can use to build, train, and run ML models to potentially predict future crop yields based on climate or weather data. Though the current iteration of PYCS is not based on time series, and therefore does not forecast future yields or trends, that is the ultimate goal for the application. However, the current iteration is a potentially useful what-if tool. For example, a farmer could design worst-case and best-case scenarios and input these hypothetical features into PYCS to create what-if predictions. This could help the farmer understand how large or small their yields might be, so they could prepare in advance to handle good, fair, and bad years appropriately and manage their resources wisely.

Alfalfa is so crucial to food security that many U.S. universities support initiatives to research this crop and publish data on the experimental growth, cultivation and harvesting of available varieties of alfalfa. These variety trial reports often include crop yield data for multiple cuts each year, contributing to the richness of the nationally available yield data. Very recent work by [7] highlights the relevance of alfalfa research, as their data come from the same variety trial reports as ours. They focus on the effects of rain on yield gaps and, like us, primarily look at rainfed rather than irrigated sites [7]. The importance of alfalfa and this abundance of alfalfa data, along with recent strides forward in ML, motivate our team to apply ML to the problem of estimating and predicting crop yields, with a focus on the alfalfa crop. As in our team's previous work [1], we use aggregated climate and yield data detailed in Tables 1 and 2.

**Table 1.** Shows the five features used to train all KY and GA models plus year harvested and class, assigned according to standard deviation and yield; these are the 7 records in the very small GA target dataset.

| Year | Yield (tons/acre) | Class | Solar Radiation (MJ/m$^2$) | Total Rainfall (mm) | Avg Min Temp (°C) | Avg Max Temp (°C) | Avg Soil Moisture (%) |
|------|-------------------|-------|----------------------------|---------------------|-------------------|-------------------|-----------------------|
| 2009 | 2.40 | 0 | 3400.10 | 740.67 | 16.10 | 27.88 | 0.19 |
| 2008 | 3.33 | 1 | 3810.06 | 664.88 | 18.44 | 29.26 | 0.12 |
| 2008 | 3.35 | 1 | 3545.32 | 413.28 | 12.54 | 24.64 | 0.13 |
| 2008 | 5.20 | 1 | 4463.92 | 599.97 | 15.77 | 28.37 | 0.19 |
| 2009 | 5.92 | 1 | 5320.75 | 1323.90 | 15.69 | 26.19 | 0.14 |
| 2009 | 6.26 | 2 | 3915.63 | 925.56 | 16.31 | 27.22 | 0.13 |
| 2010 | 6.50 | 2 | 4092.75 | 847.82 | 16.28 | 28.33 | 0.13 |

Mitigating the effects of climate change is another motivator behind this work. Evidence suggests that traditional approaches to forecasting crop yields are becoming less reliable as the Earth's climate becomes less predictable [8–15]. As the global effort to combat climate change continues, led in part by the United Nations [16], we hope this work is a positive force toward these efforts.

This work is part of an ongoing project that explores the benefits of using ML to predict biomass yields for crops. We focus on the alfalfa crop, but this research could be expanded to other crops. Our first paper concentrates on feature selection, as it presents results from a battery of tests of the following feature selection techniques: correlation-based feature selection (CFS), ReliefF method, and a wrapper method. The data for that paper is a curated mixture of alfalfa biomass yields from variety trials in Georgia (GA) and Kentucky

(KY) combined with weather and soil moisture sourced from the National Oceanic and Atmospheric Association (NOAA) [1].

**Table 2.** Shows the four features used to train all SD and OH models plus year harvested and class, assigned according to standard deviation and yield; these are the 10 records in one very small OH target dataset.

| Year | Annual Avg Yield (tons/acre) | Annual Avg Min Temp (°C) | Annual Avg Max Temp (°C) | Total Accumulated Rain (mm) | Total Accumulated Radiation (W/m$^2$) | Class |
|------|------|------|------|------|------|------|
| 2019 | 1.36 | 7.56 | 17.82 | 7462.16 | 552,381.90 | 1 |
| 2019 | 1.54 | 6.32 | 16.84 | 13,854.04 | 1,009,929.62 | 1 |
| 2010 | 1.80 | 6.76 | 17.44 | 3764.83 | 561,106.56 | 2 |
| 2010 | 1.24 | 5.88 | 16.98 | 9814.53 | 1,054,700.52 | 0 |
| 2011 | 1.28 | 6.16 | 17.26 | 6155.62 | 574,282.91 | 0 |
| 2011 | 1.89 | 6.19 | 16.88 | 7295.27 | 1,033,772.80 | 2 |
| 2011 | 0.88 | 5.74 | 16.81 | 15,084.74 | 1,518,744.53 | 0 |
| 2012 | 1.30 | 6.31 | 17.48 | 10,412.91 | 1,014,862.70 | 0 |
| 2012 | 1.57 | 6.37 | 17.09 | 11,656.76 | 1,459,165.91 | 1 |
| 2012 | 1.59 | 5.10 | 16.46 | 4568.04 | 400,766.84 | 1 |

Our second paper focuses on comparing the ML models themselves, presenting R, $R^2$, and MAE scores for most of the same models in the current and previous works. That paper expands the dataset to include the U.S. states of Pennsylvania (PA), Wisconsin (WI), and Mississippi (MS). Those results show that larger datasets lead to higher accuracies when the states are all combined into one large dataset. That work also revealed that while data from multiple sites can be combined to train one model with good $R^2$ scores above 0.90, training a model with strictly non-local data to estimate yields in a separate locality did not produce usable results with regression models [17]. This motivated us to reframe this as a classification problem in the current work, and the results are much better.

We reframed our regression problem from [17] as a classification problem with three tiers for yields: high, medium, and low. We swapped the regression models in our original application programmer interface (API) from [1,17] for analogous classification models from ScikitLearn [18]. For this work, as we are interested in estimating very small target datasets, we reduce the original target GA dataset in [1] to one alfalfa variety per year, and we annualize the cuts as detailed in Section 3, resulting in only seven records detailed in Table 1. We repeat this procedure for Ohio (OH) and South Dakota (SD), producing very small OH target datasets of 8 and 10 records and producing very small datasets for one SD town, also of 8 and 10 records. Table 2 shows our annualized, one-variety per year target data from OH with its four features. OH's features are slightly different from GA's and KY's in that OH lacks soil moisture data. OH and SD follow the same schema. In Tables 1 and 2, class labels 0, 1, and 2 correspond to low, medium, and high yields, respectively.

Though CTGAN consistently outperforms TVAE in [5], TVAE was competitive in the current work, especially when combined with XGBoost in the SNLT pipeline. The synthetic datasets generated by CTGAN train our most accurate predictor when training with KY and classifying GA, but those generated by TVAE beat CTGAN in our SD and OH experiments. SNLT trained with TVAE data produced highest accuracies of 75.0% and 70.0% using random forest (RF) and XGBoost models. SNLT trained with CTGAN data, however, produces the highest accuracy of 87.5%. Other accuracies using synthetic datasets achieved modest accuracies of 62.5% and 60.0%, but this is a significant improvement over local-only and non-local training. We determined through experimentation that accuracies increased as sample sizes increased up to 1000 or 2000, but we noticed diminishing returns after that on all models except XGBoost, so we chose 1000 or 2000 samples for all other

models. For XGBoost, which trains far faster than all the other models, we experimented with synthetic sample sizes up to 200,000, though 5000 usually delivered similar accuracies faster; therefore, all but one of our XGBoost models generate 5000 samples, and the other generates 200,000, as summarized in Section 4's results tables. Overall, non-local and local training result in low classification accuracies, with no clear winner between the two; however, our SNLT accuracies are significantly higher than non-local training or local training.

## 2. Related Work

While results in this team's previous papers are very promising and show very high accuracies using rich datasets collected by scientists and researchers, the current work considers the problem of very small datasets potentially collected by farmers at real-world farms. In previous experiments, our high accuracies are aided by the coexistence of many varieties of alfalfa growing in the same location, providing plenty of data to train and predict locally or on datasets of combined locations. On the other hand, the current work's end goal is to create a practical application that is useful to actual farmers, and we contend that farms in the real world will typically have much smaller datasets. This presents a problem where strong, accurate models are difficult or impossible to train. As a solution, PYCS offers the option to use already-trained models to estimate users' yields, even if their datasets are very small.

Our previous work has detailed some of the related work in the domain of ML and precision agriculture. Overall, the literature shows many complex techniques involving remote sensors, unmanned aerial vehicles (UAVs) or drones, computer vision, and image processing to develop vegetation indices to measure crop health [1,17]. Our work diverges from this trend in that our data come from public sources and simple weather data. More recently, Baral et al. at Kansas State University published a study of the effects of rain gaps on alfalfa yields across the United States. They aggregated alfalfa yield and weather data for 10 states, and like the current work, they obtained their yield data from university variety trials [7]. This helped inform the current work's use of data from South Dakota (SD) State University and Ohio (OH) State University, and as [7] is only interested in rainfed alfalfa, it provides a helpful guide of rainfed locations for our team to use in future experiments. The exhaustiveness of [7] inspires us to explore as many states as possible. Furthermore, they measure alfalfa yield over growing degree days (GDD), which means they do not consider days where the temperature did not meet some threshold alfalfa requires to grow, so that concept may help inform our future work [7]. Otherwise, the current work bares little similarity to [7]. That work essentially analyzed the cost in alfalfa of dry periods (rain gaps) using frontier function analysis and boundary function analysis. They also used conditional inference trees (CITs) to determine which weather features most affect crop yields. They did not compare ML models' ability to predict or estimate alfalfa yields, and they do not mention a goal of forecasting future yields using time series and ML, as the current work does.

The current work takes advantage of one DL model that has shown great promise in recent years—the generative adversarial network (GAN). Training a GAN requires finding the Nash equilibrium in a minimax game, an idea which precipitated a breakthrough in economics in the 20th century [19]. The GAN applies game theory to DL training, as a competition between two entities drives up the performance of each. One entity, called the generator, creates simulated, fictitious data that resembles the real training data. The other entity, called the discriminator, looks at real and simulated data and tries to determine which are which. As the discriminator improves at guessing, this causes the generator to generate more convincing fakes, which causes the discriminator to continue to become a better guesser, and so on. This equates to a game where each player maximizes its own chances of winning by minimizing its opponent's chances, also known as a minimax game [4]. Though mainly leading to breakthroughs in the computer vision domain, GANs have also shown promise in other applications such as differential privacy with DPGAN [20],

generating private tabular health record data with medical GAN (MedGAN) [21], and synthesizing table data with TableGAN in [22]. Refs. [20–22] demonstrate a GAN's ability to generate data that is statistically similar to the training data but introduces enough noise that connections between real people and the data are broken, protecting individuals' privacy. MedGAN combines a GAN with an autoencoder to apply this concept to health records, where privacy is a clear concern that may inhibit individuals' willingness to share data, but also where abundant data can make ML more useful in helping cure illness [21].

CTGAN is called a conditional GAN because it uses a conditional generator and training-by-sampling to address class imbalance in discrete columns. CTGAN introduces a technique called "mode-specific normalization" to normalize columns individually, and its underlying networks are fully connected [5].

We also investigate the usefulness of variational autoencoders (VAEs) in synthesizing table data, as VAEs are another class of generative neural network that has received a lot of attention and shown promising results in recent years [5]. Specifically, this work uses the tabular variational autoencoder (TVAE) proposed in [5], which adapts a plain VAE to address distribution concerns particular to generating table data.

Another somewhat recent development that helped our pipeline produce some of its best results is extreme gradient boosting or the XGBoost model. Generally, boosting combines an ensemble of weak learners to create one strong learner. One popular boosting algorithm called Adaboost often works well with random forest (RF) or decision tree (DT) models trained iteratively, where each iteration improves on the errors of the previous by weighting incorrect predictions higher in subsequent iterations, until some predefined depth or maximum accuracy is reached [23]. In XGBoost, DTs are trained to minimize prediction errors using gradient descent, creating an optimized strong learner, and this results in extremely fast training times, facilitating training on very large datasets [24].

## 3. Materials and Methods

Our alfalfa yield data come from publicly available alfalfa variety trials conducted by the University of Georgia [25], University of Kentucky [26], South Dakota State University [27], and Ohio State University [28]. Our weather data in GA and KY come from various NOAA resources, while our weather data for SD and OH comes from the Daymet tool, provided by Oak Ridge National Laboratory in Oak Ridge, TN and supported by NASA [29]. We provide our curated, aggregated datasets along with source code at the Github repository for this project: www.github.com/thejonathanvancetrance/SNLT (accessed on 29 November 2022). Every record in the original KY and GA datasets include the following 15 features: State, City, Date Sown, Variety, Date of Cut, Julian Day, Yield (tons/acre), Time Since Sown (Days), Time Since Last Harvest (Days), Total Radiation (MJ/m$^2$), Total Rainfall (mm), Avg Air Temp (°C), Avg Min Temp (°C), Avg Max Temp (°C), Avg Soil Moisture (%). Table 1 depicts the five most salient features we settled on for training for KY and GA experiments, minus class, year, and yield: average minimum temperature, average maximum temperature, total rainfall, solar radiation, and soil moisture. In SD and OH experiments, we omitted soil moisture, because we cannot consistently obtain soil moisture data for most locations as explained in Section 5. We set the thresholds of our yield classes according to standard deviation, where we label yields less than −1 standard deviations below the mean as low (class 1), yields between −1 and +1 standard deviations as medium (class 2), and yields more than +1 standard deviations above the mean as high yield in GA and KY. Our yield thresholds for all locations are summarized in Table 3. We settled on the hyperparameters shown in Appendix A via experimentation.

**Table 3.** We calculated the standard deviations (stdvs) of the yields to assign class labels for yields in each location; we used three different thresholds (1, 0.8, 0.5) to avoid high class imbalance.

| Location | Class 1 (Low Yield) | Class 2 (Medium Yield) | Class 3 (High Yield) |
|---|---|---|---|
| KY & GA | <−1 stdvs below mean | −1 < +1 stdvs below/above mean | >+1 stdvs above mean |
| SD & OH | <−0.5 stdvs below mean | −0.5 < +0.5 stdvs below/above mean | >+0.5 stdvs above mean |
| SD & Highmore, SD | <−0.8 stdvs below mean | −0.8 < +0.8 stdvs below/above mean | >+0.8 stdvs above mean |

We began the current work with the same GA and KY data, plus the code base and API that we used in [1,17]. We reduced the target GA dataset to seven records by including only one variety per year and annualizing the data. A real-world farm's dataset may not include multiple varieties, so we attempt to simulate this in our target datasets. Additionally, annualizing our data helps reduce the influence of short-term anomalous weather conditions on our models. Furthermore, some variety trial data, such as PA data, are only reported annually, so some of our data were originally annualized to be compatible with those. We expect this annualized data to simulate accuracies we might expect from PYCS when using already-trained models to estimate very small datasets. In GA and KY, the alfalfa yields are total annual yields. As this project is ongoing and evolving, we used total yields originally to match data from other states that report annually. As we expanded our data to include OH and SD, we decided that the average yield from all cuts in a year may be more informative when multiple cut information is available, mainly because some years have more cuts than others, potentially inflating the total annual yield. The current work is intended as a step toward forecasting future yields, but we are not truly forecasting yet, as we train with data from the entire year up to the date of the final cut in GA and KY. Therefore, our total rainfall, total solar radiation, average minimum and maximum temperature, and average soil moisture data typically represent about the first eight or nine months of that year in GA and KY. In SD and OH, those features represent the totals and averages over the lifetime of one planting, which may be several years, up to the last cut of that year. Table 1 lists "Solar Radiation" and "Total Rainfall" while Table 2 lists "Total Accumulated Radiation" and "Total Accumulated Rain" because of these differing approaches, where Table 1 reflects one year's weather until the final cut, and Table 2 reflects weather totals over the lifetime of that planting up to the final cut of that year. The average minimum and maximum daily temperatures are sums of each day's minimum and sum of each day's maximum temperatures, divided by the number of days considered. Table 1 weather data is sourced from the NOAA, which reports solar radiation as solar irradiation in $MJ/m^2$, while Table 2 weather data is sourced from Daymet, which reports solar radiation as solar irradiance in $W/m^2$. Though Tables 1 and 2 reflect slightly different annualization processes, we think both are reasonable approaches.

The annualization process reflected in Table 2 resulted in a target dataset of 10 records of OH data, then we removed two arbitrary records to create an eight-record dataset for better comparison with GA results. Once again, we followed this process with SD data, creating a local target ten-record dataset of data from only Highmore, SD, leaving the other SD locations for non-local training and synthesis; then, we again arbitrarily removed two records to create an eight-record Highmore dataset for comparison. We also annualized the SD training dataset.

We modified the original code base to build classification models instead of their regression counterparts, using the ScikitLearn [18] ML library for the Python programming language along with NumPy [30]. We swapped [18]'s k-nearest neighbors (KNN) regressor for a KNN classifier, random forest (RF) regressor for a RF classifier, linear regression for logistic regression (LR), support vector regressor for a support vector classifier (SVC), decision tree regressor for decision tree classifier (DT), and the multilayer perceptron (MLP) regressor for an MLP classifier (which we refer as the artificial neural network or ANN interchangeably). We added ScikitLearn's XGBoost classifier, not tried in our previous

work, to our cast of models. Since the task is now one of classification, we modified the API used in [1] to provide accuracy percentage scores indicating percent of estimates correct, rather than correlation coefficients or mean absolute errors (MAEs). We used Matplotlib to generate confusion matrices [31].

For local training, we trained all models on our very small GA, OH, and SD datasets reflected in Tables 1 and 2, then we used those models to classify those same datasets, resulting in low accuracies as shown in Tables 4–8. For regular non-local training, we trained all models on: the KY dataset with 183 records from Lexington, KY; the complete SD dataset with 767 records; and the SD dataset minus Highmore with 604 records. Next, we used those models to classify the local GA, OH, and Highmore, SD datasets, resulting in the different but mostly unimproved accuracies shown in Tables 4–8.

Our third and featured set of experiments explored our SNLT pipeline, depicted in Figure 1. SNLT begins by choosing a dataset that is not local to the target area that is being classified as high, medium, or low yield. That non-local dataset is fed to a data synthesizer, which in the case of the current work is either CTGAN or TVAE. Figure 1 depicts SNLT using a fully connected deep CTGAN, but other synthesis techniques could be used in place of this in the SNLT pipeline. The synthesizer outputs a larger training dataset than the original non-local dataset, and the size of the new dataset is an adjustable parameter of SNLT. As our experiments confirmed, these larger synthesized datasets usually train more accurate classifiers than those trained by the non-local data. Next, a model is selected for training with the synthesized data. We experimented with ANN, KNN, RF, DT, SVC, LR, and XGBoost, and since DT and XGBoost produced the best results, Figure 1 depicts SNLT as using a tree-style model, though the others are also valid. Finally, the trained model is used to classify the target dataset's alfalfa yield. This dataset would typically be too small to adequately train most models or effectively split into test and training. Based on sun, rain, temperature, and soil moisture when available, the model classifies target alfalfa yields as low, medium, or high, and SNLT produces accuracy scores as compared to the true class labels.

Our experiments begin with the same labeled KY and SD non-local training data as before, but we feed them through a generative model that synthesizes the data, resulting in larger datasets of 1000 to 200,000 records. The generative model in the pipeline is interchangeable with any generative model or synthesis technique, but the current work explores CTGAN and TVAE [5]. Next, we use these synthetic source states' data to train all our models, and we again use those trained models to classify labels for our small GA, OH, and SD targets, resulting in the promising accuracies depicted in Tables 4–8.

**Table 4.** Train KY, test GA: best accuracies from GA local training, non-local training (KY), and SNLT with CTGAN and TVAE (KY), each for three class labels (high, medium, and low yield), with synthetic dataset size of 1000 samples, except XGBoost, which is 200,000 samples. Highest accuracy is bold.

| Model | Local | Non-Local | SNLT (TVAE) | SNLT (CTGAN) |
|-------|-------|-----------|-------------|--------------|
| ANN | 28.6% | 42.8% | 57.1% | 57.1% |
| KNN | 28.6% | 28.6% | 42.8% | 57.1% |
| LR | NA * | 14.3% | 14.3% | 57.1% |
| DT | 42.8% | 28.6% | 57.1% | **85.7%** |
| SVC | 14.3% | 14.3% | 57.1% | 57.1% |
| RF | 28.6% | 42.8% | 42.8% | 57.1% |
| XGB | 28.6% | 28.6% | 28.6% | 57.1% |

* failed to train, omitted.

**Table 5.** Train SD, test OH with 8 samples: best accuracies (bold) from OH local training, non-local training (SD), and SNLT with CTGAN and TVAE (SD), each for three class labels (high, medium, and low yield), plus size of synthesized dataset.

| Model | Local | Non-Local | SNLT (TVAE) | SNLT (CTGAN) | Sample Size |
|-------|-------|-----------|-------------|--------------|-------------|
| ANN | 25.0% | 12.5% | 62.5% | 62.5% | 5000 |
| KNN | 25.0% | 12.5% | 62.5% | 50.0% | 2000 |
| LR | 25.0% | NA * | 25.0% | 50.0% | 2000 |
| DT | 37.5% | 12.5% | 62.5% | 62.5% | 3000 |
| SVC | 25.0% | 12.5% | 50.0% | 50.0% | 2000 |
| RF | 12.5% | 12.5% | **75.0%** | 62.5% | 2000 |
| XGB | NA * | 50.0% | **75.0%** | 62.5% | 5000 |

* failed to train, omitted.

**Table 6.** Train SD, test OH with 10 samples: best accuracies (bold) from OH local training, non-local training (SD), and SNLT with CTGAN and TVAE (SD), each for three class labels (high, medium, and low yield), plus size of synthesized dataset.

| Model | Local | Non-Local | SNLT (TVAE) | SNLT (CTGAN) | Sample Size |
|-------|-------|-----------|-------------|--------------|-------------|
| ANN | 20.0% | 10.0% | 50.0% | 40.0% | 2000 |
| KNN | 10.0% | 40.0% | 50.0% | 40.0% | 2000 |
| LR | 10.0% | 10.0% | 30.0% | 40.0% | 2000 |
| DT | 10.0% | 20.0% | 50.0% | 50.0% | 2000 |
| SVC | 20.0% | 20.0% | 40.0% | 60.0% | 2000 |
| RF | 30.0% | 30.0% | 50.0% | 50.0% | 2000 |
| XGB | 10.0% | 50.0% | **70.0%** | 60.0% | 5000 |

**Table 7.** Train SD, test Highmore, SD only one variety per year resulting in 10 target samples: best accuracies (bold) from Highmore local training, non-local training (rest of SD), and SNLT with CTGAN and TVAE (rest of SD), each for three class labels (high, medium, and low yield), plus size of synthesized dataset.

| Model | Local | Non-Local | SNLT (TVAE) | SNLT (CTGAN) | Sample Size |
|-------|-------|-----------|-------------|--------------|-------------|
| ANN | 20.0% | 60.0% | 50.0% | 50.0% | 2000 |
| KNN | 40.0% | 40.0% | 50.0% | 50.0% | 2000 |
| LR | 30.0% | 50.0% | 50.0% | 60.0% | 2000 |
| DT | 30.0% | 50.0% | 60.0% | 50.0% | 2000 |
| SVC | 40.0% | 50.0% | 60.0% | 50.0% | 2000 |
| RF | 20.0% | 60.0% | 60.0% | 50.0% | 2000 |
| XGB | 30.0% | 60.0% | **70.0%** | 50.0% | 5000 |

**Table 8.** Train SD, test Highmore, SD only one variety per year and 8 target samples: best accuracies (bold) from Highmore local training, non-local training (rest of SD), and SNLT with CTGAN and TVAE (rest of SD), each for three class labels (high, medium, and low yield), plus size of synthesized dataset.

| Model | Local | Non-Local | SNLT (TVAE) | SNLT (CTGAN) | Sample Size |
|---|---|---|---|---|---|
| ANN | 25.0% | 37.5% | 62.5% | 50.0% | 2000 |
| KNN | 12.5% | 50.0% | 50.0% | 50.0% | 2000 |
| LR | 25.0% | 50.0% | 50.0% | 50.0% | 2000 |
| DT | 12.5% | 25.0% | 62.5% | 50.0% | 2000 |
| SVC | 37.5% | 37.5% | 62.5% | 50.0% | 2000 |
| RF | 12.5% | 37.5% | 62.5% | 50.0% | 2000 |
| XGB | 25.0% | 50.0% | **75.0%** | **75.0%** | 5000 |

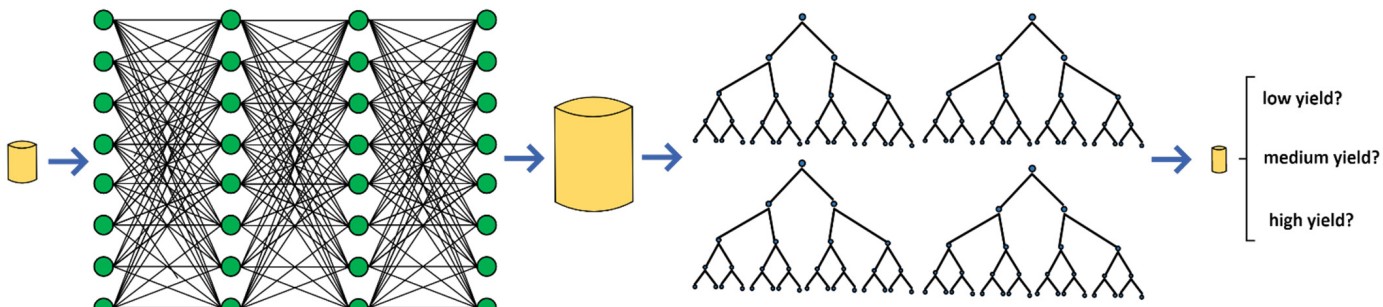

**Figure 1.** The SNLT Pipeline. Original data fed to a data synthesizer like CTGAN, which outputs lager dataset. Synthesized dataset used to train models like XGBoost. Trained model classifies very small target dataset.

## 4. Results

Whereas we previously found non-local training unviable for regression [17], our classification version of non-local training resulted in meaningful, though low accuracies which do not always beat local training, as indicated in Tables 4–8. Our SNLT pipeline produced the highest accuracy score of 85.7% (Table 4) out of these three general approaches, training a DT model with 1000 samples of data synthesized from KY using CTGAN, and classifying a very small target GA dataset of seven records. Figure 2 depicts these results as a confusion matrix. On other states' datasets of sizes 8 and 10 samples, SNLT achieved top accuracies of 75.0% and 70.0%, respectively. As Table 5 shows, SNLT achieved 75.0% accuracy training on data synthesized from SD and classifying OH, training RF and XGBoost models with 2000 and 5000 records, respectively, using TVAE, and classifying a very small target OH dataset of eight records. Figure 3 depicts these results as a confusion matrix. As Table 6 shows, SNLT achieved 70.0% accuracy training on data synthesized from SD and classifying OH, training an XGBoost model with 5000 records, using TVAE, and classifying a very small target OH dataset of ten records. As Table 7 shows, SNLT achieved 70.0% accuracy training on data synthesized from most of SD and classifying Highmore, SD, training an XGBoost model with 5000 records, using TVAE, and classifying a very small target Highmore, SD dataset of ten records. Figure 4 depicts these results as a confusion matrix. Finally, as Table 8 shows, SNLT achieved 75.0% accuracy training on data synthesized from most of SD and classifying Highmore, SD, training an XGBoost model with 5000 records, using TVAE and CTGAN, and classifying a very small target Highmore, SD dataset of eight records. Even when SNLT resulted in underwhelming accuracies of 60.0% and 62.5%, it beat regular non-local training and local training soundly and consistently as indicated in Tables 4–8. We were pleased to see XGBoost surface as a consistent top performer, as this was the only model in our cast that would train in an

acceptable timeframe on tens to hundreds of thousands of records, and the only model that did not drop in accuracy with larger sample sizes. Almost everywhere our results indicate good scores using SNLT with XGBoost using 5000 samples, we received the same results using 100,000 and 200,000 samples, though we emphasize the smaller sample sizes of 5000 because they are effectively cheaper. Each run of SNLT produces a new synthetic dataset and a new trained model, and while not all runs are equally successful, once we produce a relatively successful model and dataset, we can store them for repeated classifications.

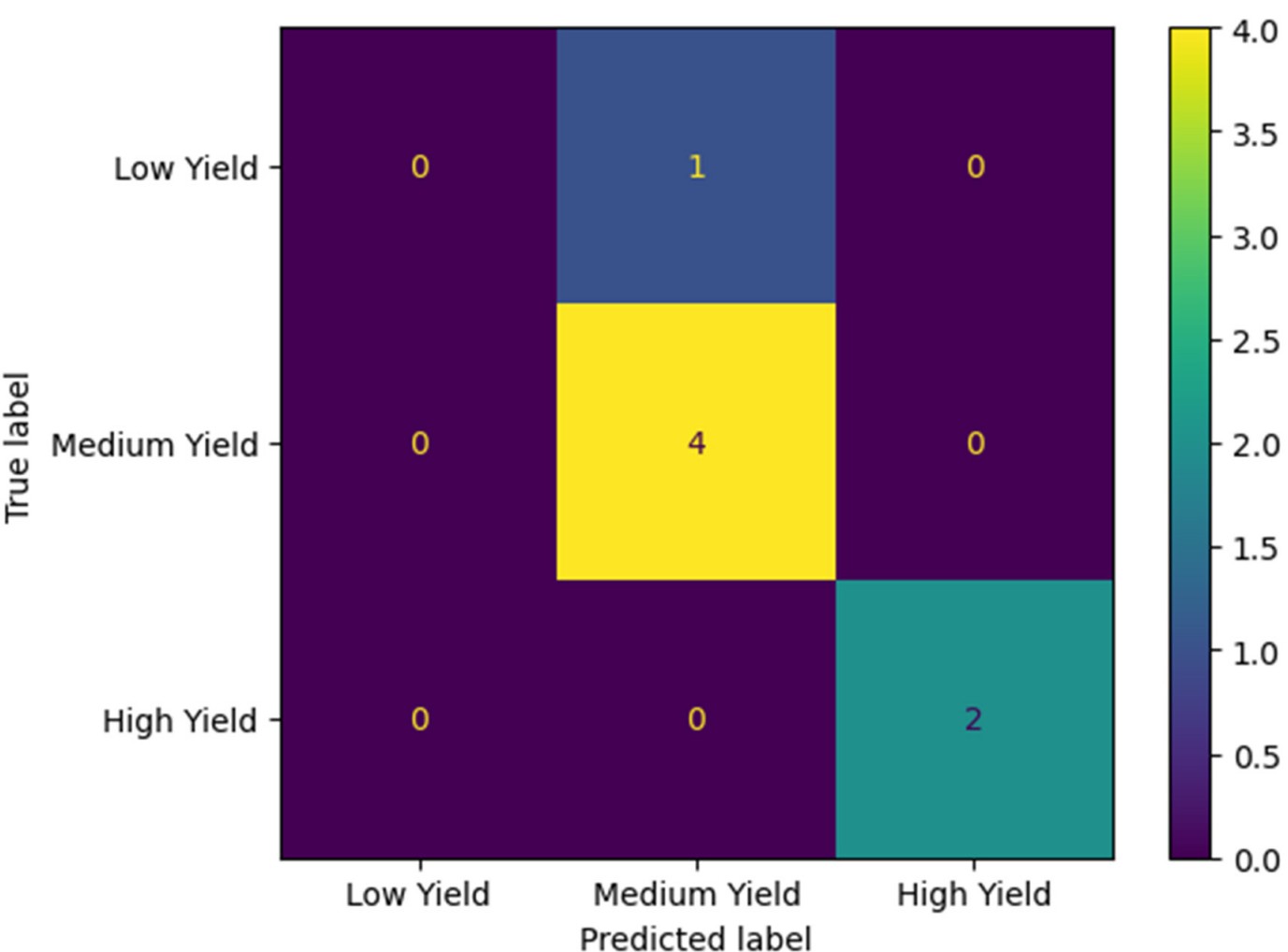

**Figure 2.** Confusion matrix for Table 4 best accuracy of 85.7% using DT and CTG with 1000 samples.

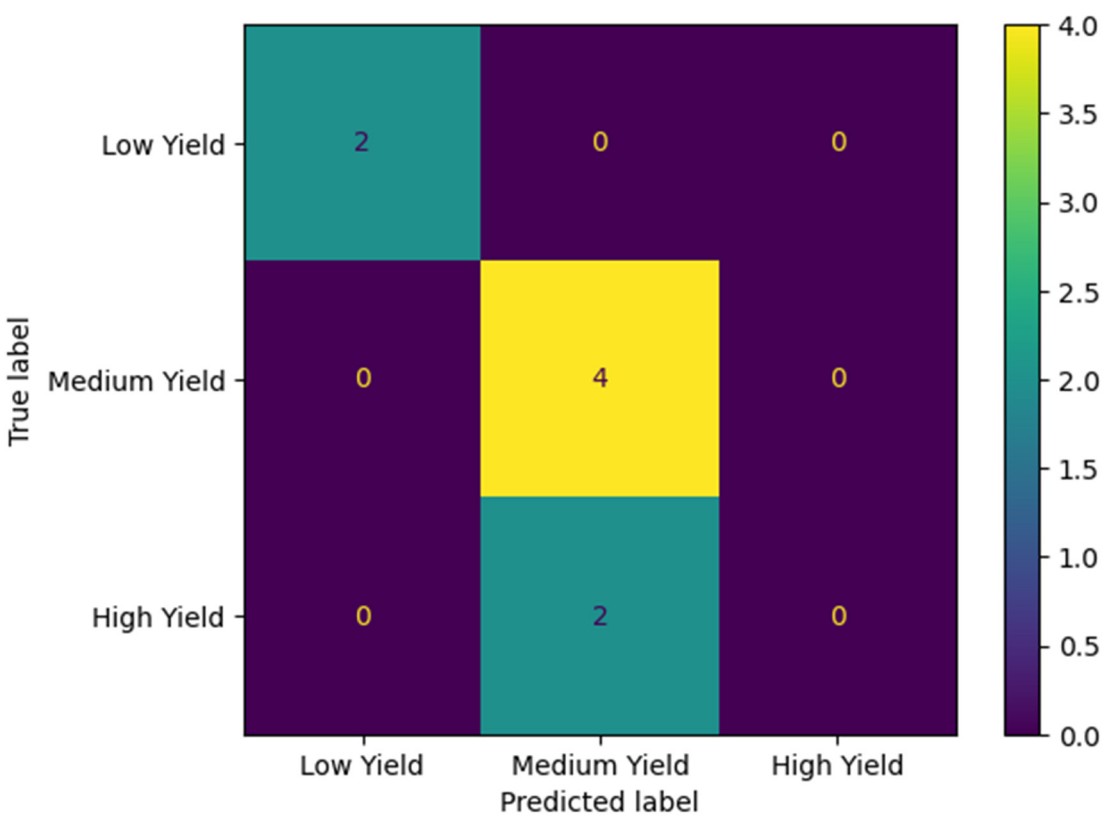

**Figure 3.** Confusion matrix for Table 5 best accuracy of 75.0% using RF and TVAE with 2000 samples.

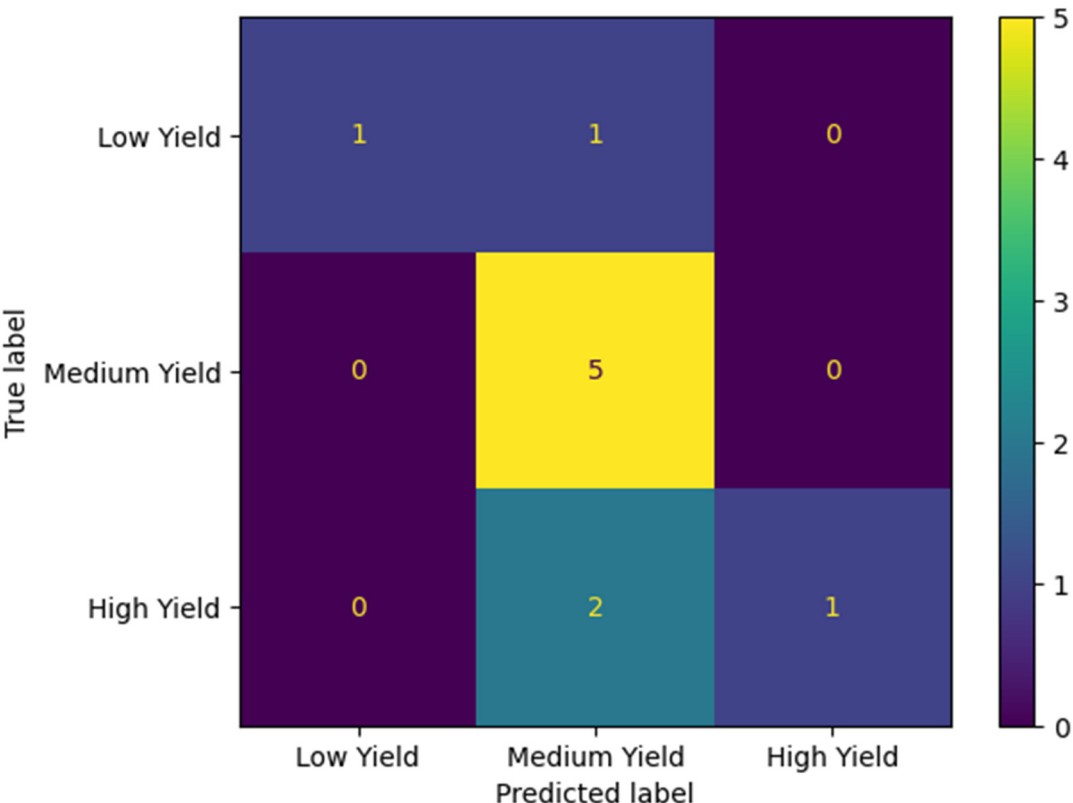

**Figure 4.** Confusion matrix for Table 7 best accuracy of 70.0% using XGB and TVAE with 5000 samples.

## 5. Discussion

Our results show that we can significantly improve estimation accuracy using our SNLT pipeline over training with non-local data or with training and estimating on the same local target dataset if it is very small. It was key for us to convert our regression problem to a classification problem. Our results are encouraging, and we think further efforts may increase our accuracies. For example, our best accuracy is higher in KY and GA experiments, where the soil moisture feature is included, and this leaves us wondering whether soil moisture is key to higher accuracies. Unfortunately, we found it difficult to obtain soil moisture data consistently throughout the U.S. Compared to our previous work, the current work aggregates yield data with much more precise weather data for SD and OH based on latitude and longitude, using the Daymet tool [30]. Before, our weather and soil moisture data came from those NOAA stations nearest to where each variety trial was conducted, which is often dozens of miles away or more. These NOAA data often include soil moisture data, but it is not always present or complete. Ideally, we would harvest data from a source that allows us to look up weather data by geolocation and would also include soil moisture data for that precise location, so we note this gap in publicly available data as well as this opportunity for someone to create such a tool.

Our results are promising enough to warrant further development of PYCS, which is still in its infancy, and the results facilitate a better understanding of how PYCS might perform and be useful in the real world. For future work, we are continuing to develop PYCS, which will provide predictions for future yields based on the past, thus providing actual crop forecasting. We have reported our best results, which come from combinations of our most accurate models with our best synthesized datasets. Our pipeline features storable models and datasets for reusability, so the live PYCS tool can estimate unseen locations using these top performers once they are created.

We are also working on gradually expanding our dataset to cover the entire contiguous United States (CONUS), or at least those where alfalfa variety trials are performed, especially where the crops are rainfed and not exclusively "Roundup Ready", as explained below. Using the Daymet [30] tool has vastly increased our efficiency with curating data and allows us to obtain weather data for any precise location where we have access to variety trials, so we expect to accelerate the growth of our datasets going forward. Since we are trying to contribute to mitigating the effects of climate change, we are more interested in rainfed crops, as rain is a climate factor, not to mention that artificial irrigation might cloud the significance of rain as a feature.

Glyphosate herbicide, known under the trade name Roundup, is a very common tool for battling pests in many crops, including alfalfa. Glyphosate kills virtually any plant it contacts, except those whose seeds are genetically engineered to be immune to glyphosate, and such modified seeds are called "Roundup-Ready". Crops grown from Roundup-Ready seeds are tolerant to glyphosate, meaning that fields where these crops are grown can be treated with the chemical, which will kill surrounding plants and weeds but not the crop itself [32]. However, in recent years, the World Health Organization determined that glyphosate is a "probable carcinogen" that may negatively affect the health of humans, animals, and our ecosystem, and that more research is necessary to determine the health consequences of such a ubiquitous product [33]. While research connecting glyphosate and related pesticides to cancer is not proven, our team generally subscribes to a non-invasive approach to precision farming, adopting the perspective that we can more effectively mitigate climate change by adapting to our environment than manipulating it. Noting this recent controversy surrounding Monsanto's Roundup herbicide, its active chemical glyphosate, and its potential carcinogenic qualities and negative environmental impacts, our work considers this product to be potentially at odds with efforts to combat climate change in a non-invasive way; therefore, our work focuses on crops that do not use glyphosate.

The next phase of our team's research will focus on true forecasting of future alfalfa yields based on time series data and models. We are exploring known time series models

like autoregressive integrated moving average (ARIMA) and vectorized autoregression (VAR), but we are also investigating a pretraining technique where we retune models already trained according to a sliding time window. We are happy to provide our data and code publicly at https://www.github.com/thejonathanvancetrance/SNLT (accessed on 29 November 2022) so that others may reproduce and extend this work.

## 6. Conclusions

The main contributions of this work are: we show that training classifiers with synthesized data often produces more accurate classifiers than training with real data, when the synthesized dataset is larger than the real one, and the target dataset is very small; we propose the novel SNLT pipeline, which trains more accurate models than local and non-local training; we propose the PYCS application, which offers a graphical user interface to perform crop yield estimation; we provide a publicly available, growing dataset of aggregated alfalfa yield and weather data. Furthermore, this work sets the stage for true forecasting of future yields, which we plan to include in future iterations of PYCS. SNLT produced the highest accuracy of 85.7% with a decision tree classifier, scored above 70% accuracy with an XGBoost classifier, and beat non-local and local training by over 40%. These results indicate that the SNLT pipeline is useful, and that it is a good feature to include in the PYCS application. When a farmer using PYCS wants to estimate alfalfa yields for a location where data is too scarce to train ML models, and when non-local training data from a nearby location is more abundant, yet too scarce to train strong models, we can synthesize larger datasets, which greatly improve the classification accuracy of our ML models. Furthermore, as researchers, we can expand the data and train and store more models as we use SNLT to conduct further experiments on new locations. Admittedly, we would like to see higher accuracies on larger target datasets with greater consistency, but we may have merely gotten our foot in the door with demonstrating this technique's usefulness. XGBoost has consistently produced encouraging results and trained very fast on very large datasets, so we plan to include it and similar models in future work, moving beyond the traditional models in our early work. A deeper study of XGBoost may reveal tuning or modifications we should perform to achieve better results. Most importantly, our results show 70% or greater accuracy in multiple locations on multiple runs, fairly well convincing us that our pipeline is useful and merits further exploration.

**Author Contributions:** Conceptualization, J.V., K.R., A.M. and F.W.M.; methodology, J.V., K.R., A.M. and F.W.M.; software, J.V.; validation, J.V., K.R., A.M. and F.W.M.; formal analysis, J.V., K.R., A.M. and F.W.M.; investigation, J.V., K.R., A.M. and F.W.M.; resources, J.V., K.R., A.M. and F.W.M.; data curation, J.V.; writing—original draft preparation, J.V.; writing—review and editing, J.V., K.R., A.M. and F.W.M.; visualization, J.V.; supervision, K.R., A.M. and F.W.M.; project administration, J.V., K.R., A.M. and F.W.M.; funding acquisition, J.V., K.R., A.M. and F.W.M. All authors have read and agreed to the published version of the manuscript.

**Funding:** This research received no external funding.

**Informed Consent Statement:** Not applicable.

**Data Availability Statement:** Our aggregated climate and alfalfa yield data from KY, GA, and WI, which we used in these experiments, and the Jupyter Notebooks [34] with already-run code and results are available at www.github.com/thejonathanvancetrance/SNLT (accessed on 29 November 2022).

**Acknowledgments:** Jonathan would like to thank his Ph.D. Committee members Hamid Arabnia and John Miller for their support and advice during this project.

**Conflicts of Interest:** The authors declare no conflict of interest.

**Appendix A**

The grid for the hyperparameters of each model is as follows:
Decision Tree

- 'criterion': ['gini'];
- 'max_depth': [5, 10, 25, 50, 100].

Random forest-

- 'n_estimators': [5, 10, 25, 50, 100];
- 'max_depth': [5, 10, 15, 20];
- 'criterion': ['gini'].

K-nearest neighbors

- 'n_neighbors': [2, 5, 10];
- 'weights': ['uniform', 'distance'];
- 'leaf_size': [5, 10, 30, 50].

Support vector classifier

- 'kernel': ['linear', 'poly', 'rbf', 'sigmoid'];
- 'C': [0.1, 1.0, 5.0, 10.0];
- 'gamma': ['scale', 'auto'];
- 'degree': [2, 3, 4, 5].

Neural Network

- 'hidden_layer_sizes': [(3), (5), (10), (3,3), (5,5), (7,7)];
- ''solver': ['sgd', 'adam'];
- 'learning_rate': ['constant', 'invscaling', 'adaptive'];
- 'learning_rate_init': [0.1, 0.01, 0.001].

Logistic Regression—default parameters; XGBoost—default parameters.

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
