# Peer review of "Data Synthesis for Alfalfa Biomass Yield Estimation"

_ai, doi:10.3390/ai4010001_

Round 1
Reviewer 1 Report
This study proposed a machine learning model-based decision tree classifier to estimate and predicate the alfalfa biomass yields. The current work is a good case study that serves the aim of this Journal. However, an in-depth analysis of the results and how the new method is constructed and evaluated must be conducted. The authors need to explain model development, SNLT Topology, and how the training is done. Also, it needs to be clarified how the model computes the proposed system efficiency compared to other studies. Besides, the technical writing has many grammatical errors, making it hard to read. The authors should justify the novelty of the proposed work as similar studies have been carried out in the existing literature.
In addition to the following:
-Enhance the conclusion to present the findings and contributions of the work.
- Add more recent literature studies that describe the works developed in the last few years and then conclude the advantages and disadvantages of each method.
-What are the current research gaps in the studies mentioned in the literature survey, and how will this work fill them?
The research paper should be written in the third person's perspective; words such as "we", "our," etc., must be avoided.
-Add more evaluation methods like MSE, RMSE, R², R²adj, MAPE, and MAE.
- Avoid using many references together, such as [8-15], etc. You should classify the studies and write a proper paragraph bout each study or category.
- Abbreviations must be written in the complete form where they are first used, such as SNLT. Check the main text and edit for the same.
- Need to add more details about the proposed SNLT pipeline.
-Too-long sentences make the meaning unclear. Consider breaking it into multiple sentences—for example, L30-L32, L38-L42, L72-L74, etc.
-Many grammatical or spelling errors make the meaning unclear, and sentence construction errors need proofreading. Improve the English language, redaction, and punctuation in general. The manuscript should undergo editing before being submitted to the Journal again.
The following are some examples:
L20-21: The highest accuracy we obtained from this pipeline results from a decision tree classifier and scored a highest accuracy of 85.7%. ….
Should be … This pipeline produced the highest accuracy of 85.7% with a decision tree classifier.
L36: on much larger datasets only performing …. Should be … on much larger datasets, only performing
L40: we turn to training our models …. Should be … we turn to train our models
L48: when used to train extreme…. Should be … when training extreme
L194: 3. Materials and Methods …. Should be … in one line, not two lines.
Reviewer 2 Report
The subject of the paper is definitely interesting for real applications in digital agriculture, and is not limited to the alfalfa case. While the accuracy provided by the proposed method marks a clear improvement with respect to existing algorithms, there is still space for further developments, also considering that the original regression problem is reformulated as a coarse classification problem with just three tiers for yields (high, medium, and low). In order to better put the method in the form of a real predictive tool to support end users, I encourage to describe how to use it in practice: in particular, how to fit it with several cuts in a year, how in advance to perform the prediction before the cut, and so on.
A list of question/remarks and minor changes to be fixed is provided:
----------------------------------------------------------------------------------
pag. 1, line 38:
"pretraining models" ---> "pretrained models"
pag. 2, line 76:
"variety trials in GA and KY" could you explain the acronyms, for non-US readers?
pag. 31, line 81:
"PA, WI, and MS" ???
pag. 2, line 92:
"we annualize the cuts" Could you explain this? How in advance does your method predict the alfalfa yelds with respect to the cut? On which time period before the cut are average temperatures, total rainfall and solar radiation measured? 12 months? less?
pag. 3, Table 1:
Averaged Min and Max temperature: are they measured day by day and then averaged over the whole year? Which year do these data refer to?
pag. 3, Table 2:
"Total Accumulated Radiation" How could it be measured in W/m2 instead of J/m2? Which is the difference between "Solar Radiation" (Table 1) and "Total Accumulated Radiation" (Table 2)? Similarly, if measured quantities in Table 1 and 2 are the same, why do they have different names? Last, which year do data in Table 2 refer to?
My suggestion: if table 1 and 2 refer to the same measured quantities, please uniform them: otherwise explain the differences. Also: Celsius degrees are neither written C nor C°, but °C
pag. 5, line 187:
"with RF or DT models" explain acronyms the first time they are met
pag. 5, lines 194-195:
"3
. Materials and Methods" Fix it
pag. 5, line 205:
"C" ---> "°C"
pag. 13, line 444:
"gini" ---> 'gini'
pag. 13, line 452:
“scale”, “auto” ---> 'scale', 'auto'
